# Circulating SPINT1 is a biomarker of pregnancies with poor placental function and fetal growth restriction

Tu'uhevaha J. Kaitu'u-Lino[1,2,7 ✉], Teresa M. MacDonald [1,2,7], Ping Cannon[1,2], Tuong-Vi Nguyen[1,2], Richard J. Hiscock[2], Nick Haan [3], Jenny E. Myers [4], Roxanne Hastie[1,2], Kirsten M. Dane[2], Anna L. Middleton[2], Intissar Bittar[5], Amanda N. Sferruzzi-Perri [6], Natasha Pritchard[1,2], Alesia Harper[1,2], Natalie J. Hannan[1,2], Valerie Kyritsis[2], Nick Crinis[5], Lisa Hui[2], Susan P. Walker[1,2,7] & Stephen Tong[1,2,7 ✉]

Placental insufficiency can cause fetal growth restriction and stillbirth. There are no reliable screening tests for placental insufficiency, especially near-term gestation when the risk of stillbirth rises. Here we show a strong association between low circulating plasma serine peptidase inhibitor Kunitz type-1 (SPINT1) concentrations at 36 weeks' gestation and low birthweight, an indicator of placental insufficiency. We generate a 4-tier risk model based on SPINT1 concentrations, where the highest risk tier has approximately a 2-5 fold risk of birthing neonates with birthweights under the 3rd, 5th, 10th and 20th centiles, whereas the lowest risk tier has a 0-0.3 fold risk. Low SPINT1 is associated with antenatal ultrasound and neonatal anthropomorphic indicators of placental insufficiency. We validate the association between low circulating SPINT1 and placental insufficiency in two other cohorts. Low circulating SPINT1 is a marker of placental insufficiency and may identify pregnancies with an elevated risk of stillbirth.

[1] Translational Obstetrics Group, The Department of Obstetrics and Gynaecology, Mercy Hospital for Women, University of Melbourne, Heidelberg 3084 Victoria, Australia. [2] Mercy Perinatal, Mercy Hospital for Women, Heidelberg 3084 Victoria, Australia. [3] Foresight Health, Adelaide, 169 Fullarton Rd., Dulwich 5065 South Australia, Australia. [4] University of Manchester, Manchester Academic Health Science Centre, St Mary's Hospital, Manchester M13 OJH, UK. [5] Pathology Department, Austin Health, Heidelberg 3084 Victoria, Australia. [6] Centre for Trophoblast Research, Department of Physiology, Development and Neuroscience, University of Cambridge, Cambridge CB2 3EG, UK. [7] These authors contributed equally: Tu'uhevaha J. Kaitu'u-Lino, Teresa M. MacDonald, Susan P. Walker, Stephen Tong. ✉email: t.klino@unimelb.edu.au; stong@unimelb.edu.au

Placental insufficiency and fetal growth restriction arise when the placenta is functioning poorly and is the largest single contributor to the risk of stillbirth. If identified antenatally, close fetal surveillance and timed birth may decrease the risk of stillbirth[1].

There is avid interest in finding better ways to detect placental insufficiency, especially among pregnancies around term gestation, because the risk of stillbirth rises sharply from around 38 weeks of pregnancy[1–3] and inducing labor is a very safe option[4,5].

The presence of placental insufficiency leads to a cascade of pathological events as the fetus tries to cope with poor placental function[6]. These include a slowing in growth resulting in fetal growth restriction, and low birth and placental weights. It can also lead to increased blood flow resistance in the uterine artery (maternal) and umbilical (placental) arteries. Decreased oxygen availability can cause the fetus to divert blood from its periphery to the brain. These blood flow changes can be measured by Doppler ultrasound of the uterine, umbilical or fetal middle cerebral artery, respectively.

When faced with decreased nutrient supply from a poorly functioning placenta, fetuses can also exhaust their energy reserves in utero leading to decreased fat stores. This can be identified postnatally by air displacement plethysmography of neonatal body composition measurements done soon after birth.

Despite these associations, antenatal tests to identify pregnancies affected by placental insufficiency perform modestly, especially at near-term gestations. The current clinical approach is to try to detect small for gestational age fetuses (SGA, fetal weight <10th centile) given they are associated with a 3–4 fold increased risk of stillbirth[1–3]. To detect SGA fetuses, it is almost universal antenatal practice to estimate the uterine size (symphysial fundal height) at clinic visits by placing a tape measure on the patient's abdomen, and selectively send those with small measurements for an ultrasound scan to estimate the fetal weight. Unfortunately, this time-honored practice of selective ultrasound scanning of women with a clinical suspicion of an SGA fetus only detects 20–30% (sensitivity) of all cases[7–9].

Furthermore, SGA itself may not be a good indicator of placental insufficiency. Many SGA fetuses are in fact constitutionally small, that is, healthy fetuses without placental insufficiency and a low baseline stillbirth risk[10]. The proportion of fetuses that are genuinely growth restricted from placental insufficiency (rather than being constitutionally small) rises sharply with decreasing fetal weight centiles below the 10th. This is reflected by an exponential increase in the incidence of stillbirth with linear step-wise decreases in birthweight centiles below the 10th. Thus, the risk of stillbirth is particularly high for fetuses with a weight <3rd centile.

Other antenatal options to identify placental insufficiency include performing ultrasounds to identify fetuses with increased umbilical artery Doppler resistance, or increased middle cerebral artery Doppler blood flow velocities. While these methods are still being evaluated, none appear likely to be good diagnostic markers of placental insufficiency, especially near term[11–13], probably because these adaptive circulatory responses do not occur with sufficient reliability to form the basis of a sensitive clinical test.

Thus, there is a clinical need to find better tests for placental insufficiency to decrease rates of stillbirth. Therefore, we set out to identify and validate new circulating markers of placental insufficiency. We prospectively collected plasma from 2003 women at 36 weeks' gestation and divided the samples to discover (cohort 1, $n = 1001$) then validate (cohort 2, $n = 1002$) biomarkers. By screening 22 proteins in cohort 1 we identify a strong association between low circulating plasma serine peptidase inhibitor, Kunitz type-1 (SPINT1) concentrations and low birthweight (<10th centile), a marker of placental insufficiency in utero. We validate this association in cohort 2 and generate a 4-tier risk model based on SPINT1 concentrations. We show SPINT1 is associated with several ultrasound and neonatal anthropomorphic parameters that are known to be associated with placental insufficiency. We also show SPINT1 expression is decreased in placentas from cases of preterm fetal growth restriction (a severe form of placental insufficiency) and from an animal model of fetal growth restriction. We validate the association between low circulating SPINT1 concentrations in a cohort from the United Kingdom and in women with preterm fetal growth restriction. Our work identifies low circulating SPINT1 as a marker of placental insufficiency.

## Results

**Screening and validation of new biomarkers.** We performed the Fetal Longitudinal Assessment of Growth study (FLAG) a prospective collection of plasma samples at 28 and 36 weeks of pregnancy. To identify biomarkers of placental insufficiency relevant to term gestation we focused on plasma samples collected around 36 weeks' gestation. We divided the cohort approximately in half to discover, then validate markers: the first 1001 (cohort 1) and the second 1002 consecutively recruited participants (cohort 2). The clinical characteristics of both cohorts are shown in Supplementary Tables 1 and 2.

To identify biomarkers of placental insufficiency we measured plasma concentrations of 22 proteins that we identified using bioinformatics as highly expressed in the placenta relative to other tissues in the body, especially on the surface of the placenta. We measured these in samples from a nested case–control subset of 105 cases of SGA (small for gestational age, or birthweight <10th centile) and 210 non-SGA controls selected from cohort 1. While the function of many of these proteins remain poorly understood, we postulated that being deliberately expressed at such high levels in the placenta, most will play important biological roles. This could then mean that the expression of some of them will be perturbed in the presence of placental insufficiency.

Circulating concentrations of 4 of the 22 proteins we screened were significantly altered with SGA compared with non-SGA controls: SPINT1, syndecan-1, growth differentiation factor-15 (GDF15), and death-associated protein kinase-1 (DAPK1, Supplementary Fig. 1). Thirteen were unchanged (Supplementary Fig. 2) and five were undetectable (listed in Supplementary Table 3). We re-assayed SPINT1 (Fig. 1a, b), syndecan-1, GDF-15 (Supplementary Fig. 3a–d) and DAPK-1 in samples from cohort 1 ($n = 997$ available) which, unlike the case–control study, brings the prevalence of SGA to that of the entire population. We confirmed that circulating concentrations of all of them except DAPK-1 (not shown) remained significantly changed.

We then validated these findings by measuring circulating SPINT1, syndecan-1 and GDF-15 at 36 weeks' gestation in cohort 2, an independent cohort from that we used to first identify their associations with SGA. All three analytes were significantly deranged in pregnancies destined to birth an SGA neonate (Fig. 1c, d shows SPINT1 in cohort 2, and Supplementary Fig. 3e–h shows GDF-15 and syndecan-1). In cohort 2 circulating placental growth factor (PlGF) was also significantly changed with SGA (Fig. 1e, f), while soluble fms-like tyrosine kinase 1 (sFlt1) concentrations were not (Supplementary Fig. 3i, j). A statistical comparison of the ROC curves showed the AUC for SPINT1 is significantly greater than PlGF ($p = 0.03$, Fig. 1g). Thus, of all these analytes, SPINT1 appeared to have the strongest association with SGA.

Circulating SPINT1 concentrations were already significantly decreased at 28 weeks among those destined to birth an SGA neonate (data derived from case-cohort set selected from

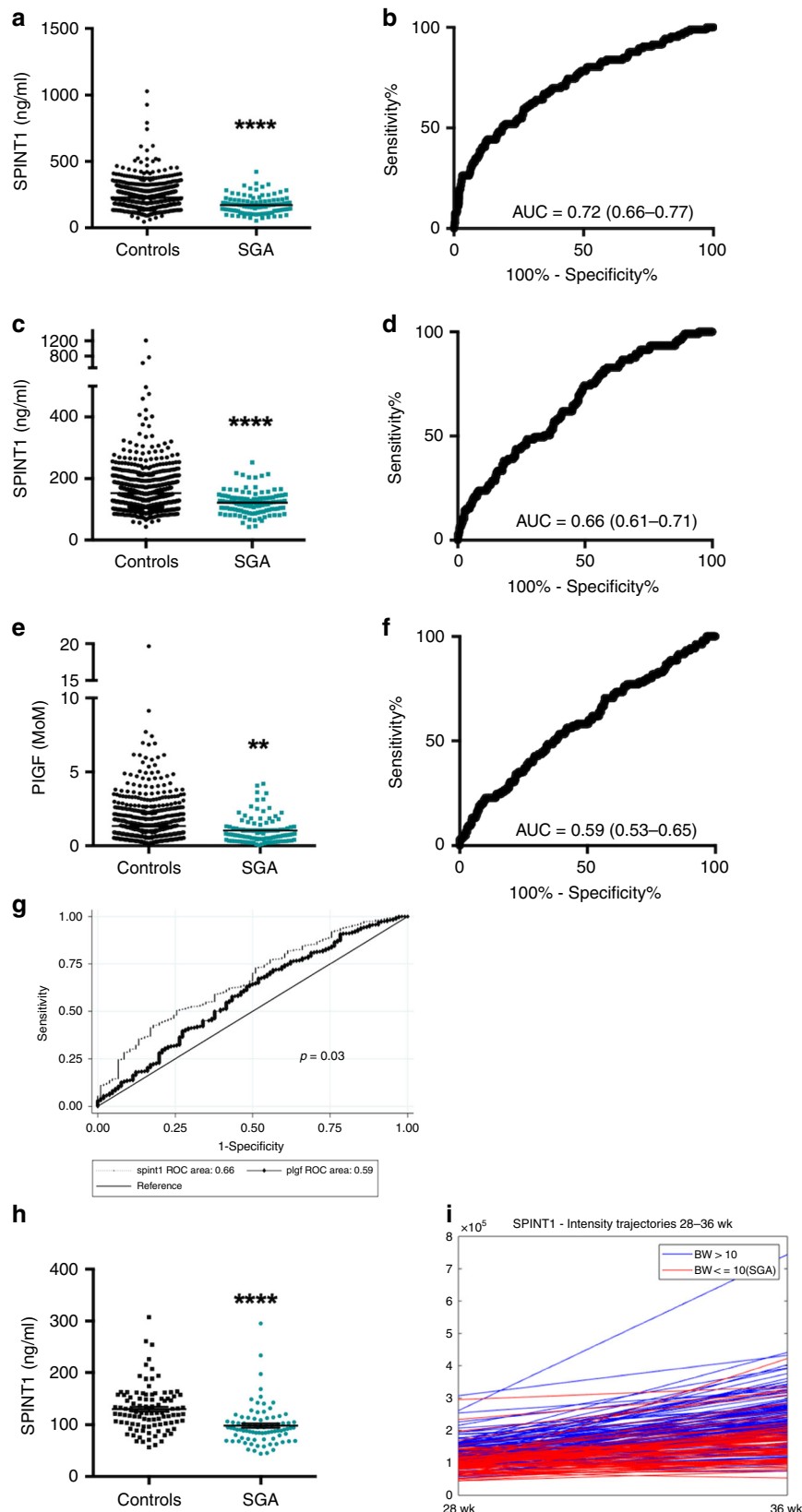

cohort 1, see Fig. 1h). For many cases of SGA where SPINT1 concentrations were low at 28 weeks' gestation, levels remained depressed at 36 weeks' gestation (Fig. 1i) suggesting levels may be low for months prior to the birth of an SGA infant at term gestation.

**Developing and validating tests to predict low birthweight.** Circulating SPINT1 concentrations at 36 weeks' gestation in cohort 2 (our independent validation cohort where SPINT1 reads were available for $n = 998$) identified a group with an elevated risk of birthing low birthweight infants (Table 1). Of note, 37.9%

**Fig. 1 Plasma concentrations of SPINT1 were significantly reduced preceding the birth of a small for gestational age (SGA) infant.** Scatter plots and receiver operator curves of SPINT1 in two cohorts (**a**, **b**; cohort 1, **c**, **d**; cohort 2) demonstrate SPINT1 concentrations were significantly reduced (**a**, $p = 2.70 \times 10^{-13}$, **b**, $p = 5.24 \times 10^{-8}$) in the circulation of women at 36 weeks who subsequently delivered an SGA infant (birthweight < 10th centile). Plasma PlGF concentrations at 36 weeks were also significantly reduced in cohort 2 (**e**, **f**, $p = 3.81 \times 10^{-3}$). A comparison between receiver operator curves for SPINT1 and PlGF indicated SPINT1 had improved area under the receiver operator curve (**g**). In a case–control sample set selected from cohort 1, SPINT1 was significantly reduced at 28 weeks' gestation (**h**, $p = 3.59 \times 10^{-9}$). Cases where SPINT1 were low at 28 weeks' gestation tended to remain low by 36 weeks' gestation (**i**). Cohort 1, $n = 891$ controls, $n = 106$ SGA. Cohort 2 $n = 893$ controls, $n = 105$ SGA, Case control at 28 weeks, $n = 100$ controls, $n = 84$ SGA. \*\*$p < 0.01$, \*\*\*\*$p < 0.0001$, groups compared using two-tailed Mann–Whitney $U$ tests. AUC—area under the ROC curve, with 95% confidence intervals given in brackets. Data depicted for **a**, **c** and **e** are mean ± s.e.m. Source data included as a Source Data file.

who screened positive had a birthweight centile <20th. This may be clinically significant given a fetal weight <20th centile still has a stillbirth risk twofold that of fetuses at around 50–90th centiles[3,14]. In contrast, the diagnostic performance of PlGF was more modest (Supplementary Table 4). Given significant differences in some maternal parameters between the groups (such as age, smoking, parity and body mass index), we also examined the predictive ability of SPINT1 at 36 weeks' gestation to detect fetuses <10th, <5th, and <3rd centiles after adjusting for these maternal factors. This did not significantly alter the performance of circulating SPINT1 (Supplementary Table 5). SPINT1 also performed better as a diagnostic test compared with combining clinical characteristics that are associated with an increased risk of birthing a low birthweight neonate (Supplementary Table 5).

The a priori intent of the FLAG study was to try to find a combination of markers providing the best diagnostic performance to detect fetal growth restriction. However, using logistic regression approaches we were unable to identify a multi-marker predictive test that added to the test performance of SPINT1 alone to identify neonates destined to be born <10th, <5th and <3rd birthweight centiles (Supplementary Tables 6, 7, 8 respectively). Given it has been previously proposed that sFlt1 and PlGF expressed as a ratio may be able to predict the presence of placental insufficiency[15,16], we also assessed the predictive performance of sFlt1/PlGF (Supplementary Table 9). Expressing sFlt1 and PlGF as a ratio produced screening tests that performed more poorly compared with either PlGF alone (Supplementary Table 4) or to SPINT1 (Table 1).

Using cohort 2, we developed a 4-tier risk model based on different SPINT1 MoM concentrations (Table 2). Those among the highest tier of risk (7.1% of cohort 2 with the lowest SPINT1 MoMs) had 14.1%, 19.7%, 28.2% and 46.5% risk of delivering neonates with birthweights <3rd, <5th, <10th and <20th centiles respectively. In contrast, those within the lowest tier of risk (9.1% of cohort 1) had 0.0%, 1.1%, 1.1% and 6.6% risk of birthing neonates at <3rd, <5th, <10th and <20th centiles. In this 4-tier risk model, rates of birthing infants at these birthweight centiles compared with the baseline population prevalence in the cohorts were elevated by 2–5 fold if circulating SPINT1 concentrations were within tier 1 (highest risk), similar to the background population prevalence for tier 2, around half of the population prevalence in tier 3, and were very low in tier 4.

**SPINT1 and clinical markers of placental insufficiency.** We next investigated the association between SPINT1 and various clinical parameters of placental insufficiency. We did this by examining data from across both cohorts 1 and 2 (after completing our discovery and validation studies).

We performed ultrasounds at 36 weeks' gestation in a subgroup of 347 nulliparous women (spread across cohorts 1 and 2), followed by air displacement plethysmography (PEA-POD) on the neonates after birth. SPINT1 was negatively correlated with the uterine artery (Fig. 2a) but not umbilical artery Doppler velocity (Supplementary Fig. 4a) measured at

36 weeks' gestation and was positively correlated with neonatal lean mass (Fig. 2b) but not fat mass (Supplementary Figure 4b,c). SPINT1 was also strongly correlated with placental weight (Fig. 2c). In contrast, the associations between PlGF and these indicators of placental insufficiency were either not significant, or more modest (Supplementary Fig. 5).

We also found a step-wise decrease in plasma SPINT1 concentrations between controls (>10th centile) and decreasing birthweight centiles (Fig. 2d). Plasma SPINT1 concentrations at 36 weeks were continuously correlated across all birthweight centiles (Fig. 2e, available data from cohorts 1 and 2 were combined).

**Validating SPINT1 in an independent high-risk cohort.** We next sought to validate the association between low circulating SPINT1 concentrations and placental insufficiency in an independent cohort. We measured plasma SPINT1 concentrations obtained from women at 24–34 weeks' gestation presenting to the Manchester Antenatal Vascular Service (MAViS clinic, clinical characteristics shown in Supplementary Table 10). In contrast to the low-risk cohort recruited in the FLAG study, women referred to this clinic were those with current hypertension, or a hypertensive disorder in a previous pregnancy. They were high-risk pregnancies because pregnant women with pre-existing vascular disease are at significantly increased risk of preeclampsia, SGA or fetal growth restriction.

Circulating SPINT1 concentrations in this cohort were significantly depressed among 83 women who birthed an SGA neonate ($n = 83$) compared with 208 who did not (Fig. 3a, b) and levels were further reduced among those who birthed neonates <5th birthweight centile (Fig. 3c). Similar to findings made in the FLAG cohort (Fig. 2e), SPINT1 concentrations were continuously correlated with birthweight centiles (Fig. 3d) and the uterine artery Doppler velocities measurements taken at the time of blood sampling (Fig. 3e). Furthermore, circulating SPINT1 concentrations at 24–34 weeks' gestation was also correlated with uterine artery Doppler velocities (Fig. 3f) and placental surface area (Fig. 3g) measured by ultrasound at 22–24 weeks, that is, ultrasound findings from the mid-trimester (suggesting the early development of placental insufficiency) which preceded blood sampling by several weeks.

Thus, we validated that circulating SPINT1 concentrations are correlated with several clinical parameters that are associated with placental insufficiency in a high-risk cohort from a different country than the cohort we used to discover the association.

**Circulating SPINT1 in severe fetal growth restriction.** Most cases of preterm fetal growth restriction requiring iatrogenic birth remote from term are affected by severe placental insufficiency. In cases of preterm fetal growth restriction (delivered < 34 weeks' gestation), placental SPINT1 mRNA and protein expression (Fig. 4a, b) were significantly reduced compared with controls (patient characteristics shown in Supplementary Table 11).

**Table 1 Diagnostic performance of circulating SPINT1 concentrations at $35^{+0}$–$37^{+0}$ weeks' gestation to identify fetuses born at varying low birthweight cut-offs.**

|  | Birthweight < 3rd centile | Birthweight < 5th centile | Birthweight < 10th centile | Birthweight < 20th centile | Birthweight < 5th centile and nursery admission |
|---|---|---|---|---|---|
| Positive predictive value | 10.2% (5.2-17.5) | 16.1% (9.8-24.2) | 22.6% (15.3-31.3) | 37.9% (29.3-47.1) | 4.8% (1.6-10.8) |
| Negative predictive value | 98.3% (97.2-99.1) | 96.5% (95.1-97.6) | 90.9% (88.9-92.8) | 78.4% (75.5- 81.1) | 99.7% (99.0-99.9) |
| Risk ratio (95% CI) | 6.04 (2.85-12.82) | 4.59 (2.66-7.93) | 2.50 (1.68-3.71) | 1.75 (1.35-2.27) | 14.2 (3.44-58.47) |
| Sensitivity | 42.3% (23.4-63.1) | 36.7% (23.4-51.7) | 24.5% (16.7-33.8) | 19.9% (15.0-25.6) | 62.5% (24.5-91.5) |
| Specificity | 90.0% (88.0-91.8) | 90.1% (88.0-91.9) | 90.0% (87.9- 91.9) | 89.9% (87.5-91.9) | 89.9% (87.9-91.7) |

This data was derived from Cohort 2, our validation cohort and calculated from the 998 participants where there was a SPINT1 readout from the assay (a further four samples did not detect SPINT1 and were excluded).

**Table 2 Four-tier risk model for delivery of neonates with different low birthweight cut-offs.**

| TIER | SPINT1 MoM cut-off | % (no) within tier | Birthweight < 3rd centile | Birthweight < 5th centile | Birthweight < 10th centile | Birthweight < 20th centile | Birthweight < 5th centile and nursery admission |
|---|---|---|---|---|---|---|---|
| 1: High risk | <0.63 | 7.1% (71) | 14.1% | 19.7% | 28.2% | 46.5% | 7.0% |
| 2: Normal risk | 0.63-1.1 | 57.0% (569) | 2.3% | 4.4% | 11.8% | 27.4% | 0.4% |
| 3: Lower risk | 1.1-1.6 | 26.8% (268) | 1.1% | 3.0% | 6.3% | 14.9% | 0.4% |
| 4: Lowest risk | >1.6 | 9.1% (91) | 0.0% | 1.1% | 1.1% | 6.6% | 0.0% |

The data were derived from cohort 2 (validation cohort). Risk model developed by selecting different thresholds of circulating levels SPINT1 MoM concentrations measured among pregnant women at $35^{+0}$-$37^{+0}$ weeks' gestation.

Circulating plasma SPINT1 concentrations were also reduced among pregnancies complicated by preterm fetal growth restriction compared with healthy pregnancies (patient characteristics given in Supplementary Table 12), with an area under the receiver operated curve of 0.948 (Fig. 4c, d). Changes in serum from the same cohort were far less pronounced (Supplementary Fig. 4d), suggesting SPINT1 should be measured in plasma, not serum.

**SPINT1 in a mouse model of fetal growth restriction.** We induced placental insufficiency and fetal growth restriction using a mouse model of late-term maternal hypoxic exposure[17]. This led to fetuses with 25% reduced fetal weight (Supplementary Fig. 6). Placental *Spint1* mRNA and SPINT1 protein expression were significantly reduced among cases where maternal hypoxia-induced fetal growth restriction (Fig. 4e, f). These data are consistent with the possibility that SPINT1 may play an important biological role in normal placental function.

## Discussion

By screening proteins that are highly expressed on the surface of the placenta we have identified a strong association between low circulating concentrations of SPINT1 at 36 weeks' gestation and placental insufficiency.

We have robustly proven the association between circulating SPINT1, and low birthweight given we validated the association in cohort 2, a sample set that is independent of the cohort (cohort 1) we used to discover SPINT1. Importantly, we externally validated this association in a cohort from Manchester UK, an independent cohort of samples from another country. We also showed circulating levels were lower in a further cohort of pregnancies complicated by preterm fetal growth restriction (a severe form of placental insufficiency).

Furthermore, we have generated tests using our validation cohort that may have clinical applicability, especially given all

existing tests perform modestly for detecting fetal growth restriction. A low SPINT1 identified a population with a significantly increased risk of delivering a neonate with a low birthweight. The positive predictive values were encouraging and potentially clinically acceptable as they ranged between 10 and 38%, depending on the birthweight centile cut-off. Many of these infants would have also been small in utero, and at increased risk of placental insufficiency and stillbirth while they remained undelivered.

Using cohort 2, we also propose a 4-tier risk model that identifies cohorts at high or low risk of birthing a neonate of low birthweight centile where presumably, it also identifies unborn fetuses at low weights and at increased risk of stillbirth. This model is able to assign a level of risk that is different from the baseline prevalence in just under half of the population who have the test (tiers 1, 3, and 4). This 4-tier model requires validation in an independent cohort.

If validated however, we posit that our 4-tier model has promising clinical value, given current approaches to detect growth restricted fetuses are so limited and an intervention exists that does not cause harm in the cases of false positives (induction of labor). Those in tier 1 may be offered a planned birth at term to reduce the risk of stillbirth and possibly perinatal morbidity. Alternately, it could be used to reassure a large number of women that their risk of having a growth restricted fetus at term is halved (tier 3) or extremely low (tier 4). As such, awaiting spontaneous labor beyond their due date may be a safe decision compared with the general population who were not tested. The test does not help triage risk for those in tier 2, which is a little over half of the population.

There is no gold standard clinical definition of placental insufficiency, which makes it challenging to develop a screening test that will be universally accepted. Thus, we examined the association between SPINT1 and many indicators of placental insufficiency. We found low SPINT1 concentrations at 36 weeks'

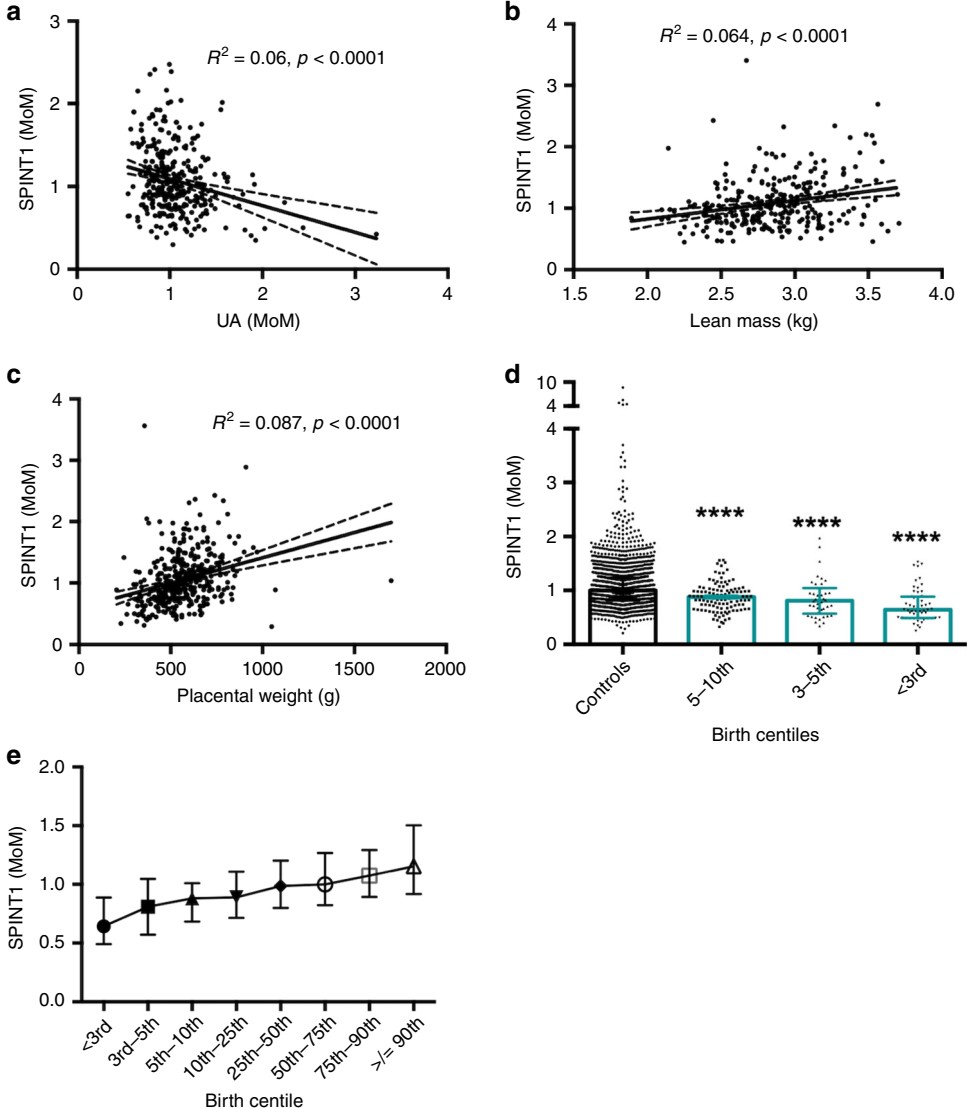

**Fig. 2 Plasma SPINT1 at 36 weeks is associated with clinical markers of placental insufficiency.** Plasma SPINT1 concentrations at 36 weeks' gestation were correlated with uterine artery (UA) Doppler flow resistance (**a**, $n = 327$), lean mass of the neonate (**b**, $n = 281$), and placental weight (**c**, $n = 305$). There was a step-wise reduction in plasma SPINT1 concentration in women whose babies were subsequently born with a birthweight below the 10th centile (**d**, $n = 1803$ controls, 5–10th $n = 115$, $p = 9.63 \times 10^{-8}$ vs control, 3–5th $n = 42$, $p = 8.57 \times 10^{-5}$ vs control, <3rd $n = 54$, $p = 3.06 \times 10^{-12}$ vs control). Plasma SPINT1 concentrations at 36 weeks were correlated with birth centiles (**e**). SPINT1 MoMs were combined for both cohort 1 and 2 and grouped according to the birth centile. Graphs **d** and **e** depict median +/− interquartile range. Each individual symbol (**a–c**) represents a patient. ****$p < 0.0001$ relative to control (two-tailed Mann–Whitney $U$ tests). Source data included as a Source Data file.

gestation (1) has a step-wise association with SGA birthweight centiles <10th, 5th, 3rd centiles, (2) is associated with reduced neonatal lean mass at birth, (3) has a negative correlation with uterine artery blood flow resistance measured at 36 weeks' gestation, and (4) is correlated with placental weight. Also, (5) SPINT1 levels are low in the placentas and the maternal circulation from pregnancies complicated by preterm fetal growth restriction (a severe form of placental insufficiency) and (6) in a mouse model of fetal growth restriction. Thus, our data suggests low circulating SPINT1 levels may reflect genuine placental insufficiency, not just low birthweight.

To induce fetal growth restriction in our murine model we induced maternal hypoxia. Our model is limited because when there is fetal growth restriction in humans the mother is not hypoxic, meaning our in vivo model does not entirely mimic the pathophysiology of this disease. Nevertheless, this model is still

likely to induce fetal hypoxia (which does occur with fetal growth restriction) and is therefore still relevant to human disease.

SPINT1 is a membrane bound cell surface protease first identified for its capacity to inhibit hepatocyte growth factor activator. Target cell surface proteases for SPINT1 include matriptase, prostasin and hepsin. Notably, *Spint1* deficiency in genetic mouse knockout studies causes embryonic lethality with severe placental abnormalities and a failure of labyrinth development[18]. Although animal studies suggest a pertinent role for SPINT1 in normal placental development, limited data has been reported on the biological role of SPINT1 in human placenta. Our interest in SPINT1 stemmed from the fact that of all tissues in the body, it appears to be most highly expressed in placenta. Now we have made the primary observation that low SPINT1 is associated with placental insufficiency, it may be worthwhile for future studies to elucidate molecular mechanisms and to tease out

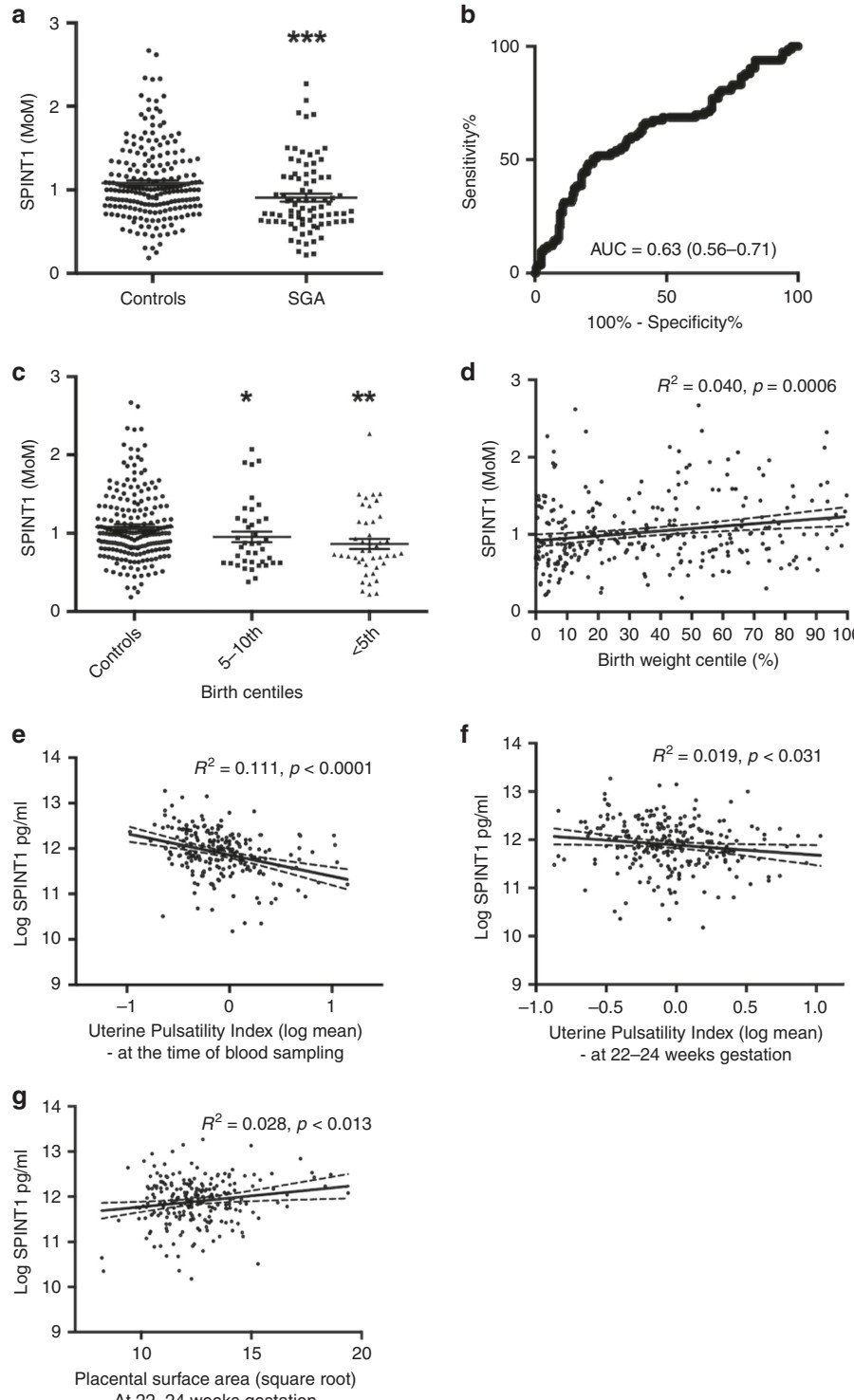

**Fig. 3 SPINT1 is reduced in an independent high-risk cohort.** Plasma SPINT1 levels were assessed in samples from women who presented at the Manchester Antenatal Vascular Service (MAViS). SPINT1 is reduced in the SGA cohort (**a**, **b**, $p = 3.31 \times 10^{-4}$), with a step-wise reduction when SGA cases were split into 5–10th and <5th centile (**c**, 5–10th $p = 2.83 \times 10^{-2}$ vs control, <5th $p = 1.10 \times 10^{-3}$ vs control). We also confirm that circulating SPINT1 correlates across all birthweight centiles (**d**). Uterine artery pulsatility index was assessed at the time of blood sampling, as well as at 22–24 weeks, and for both of these measures we demonstrate a significant correlation with circulating SPINT1 (**e**, **f**). We also confirm a significant correlation between circulating SPINT1 and estimated placental surface area at 22–24 weeks. For all panels, each individual symbol represents one patient. **a**, **b**: $n = 208$, control, $n = 83$ SGA. **c**: $n = 208$ control, $n = 40$ 5–10th, $n = 43 < 5$th. **e**: $n = 221$, **f**: $n = 244$, **g**: $n = 223$. *$p < 0.05$, **$p < 0.01$ using two-tailed Mann–Whitney $U$ tests. AUC—area under the ROC curve, with 95% confidence intervals given in brackets. Data depicted for **a** and **c** are mean ± s.e.m. Source data included as a Source Data file.

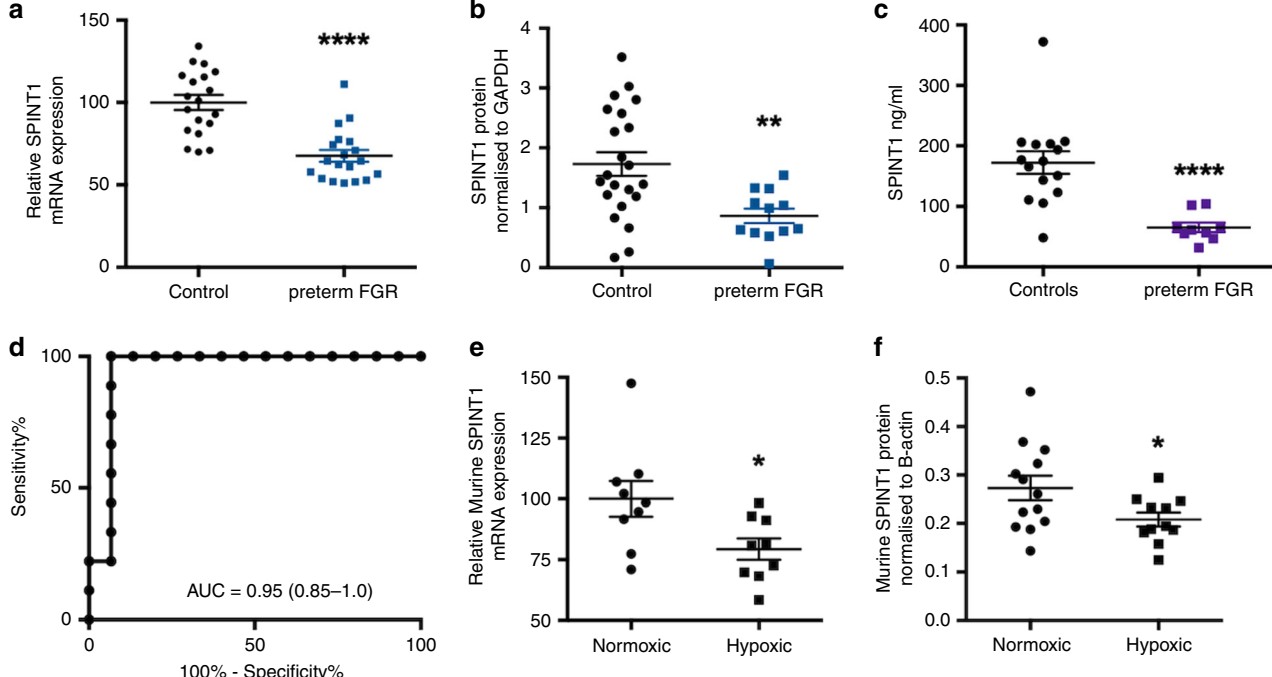

**Fig. 4 SPINT1 is reduced in preterm fetal growth restriction.** SPINT1 mRNA (**a**, $n = 19$ control, $n = 19$ fetal growth restriction - FGR) and protein expression (**b**, $n = 22$ control, $n = 12$ FGR) were significantly reduced in placentas from cases of preterm fetal growth restriction (FGR who were delivered <34 weeks' gestation and had a birthweight <10th centile) relative to gestation matched controls. Plasma SPINT1 concentrations were significantly reduced in women with preterm FGR relative to concentrations in women with healthy pregnancies, where bloods were collected at the same gestations (**c**, $n = 15$ control, $n = 9$ FGR). The area under the receiver operated curve for data shown in **c** is 0.948 (**d**). In a mouse model of hypoxia-induced fetal growth restriction, SPINT1 mRNA (**e**, $n = 9$ normoxic placentas, $n = 9$ hypoxic placentas from separate litters) and protein (**f**, $n = 13$ normoxic placentas, $n = 11$ hypoxic placentas from separate litters) expression in the placenta was significantly reduced. *$p < 0.05$, **$p < 0.01$, ***$p < 0.001$, ****$p < 0.0001$ using two-tailed Mann–Whitney $U$ tests. Individual symbols represent an individual patient, placental isolation or placenta. AUC—area under the ROC curve, with 95% confidence intervals given in brackets. Data depicted for **a**–**c**, **e**, **f** are mean ± s.e.m. Source data included as a Source Data file.

whether it is a bystander, or driver of placental disease. If the latter, then druggable therapeutic targets could be discovered within the SPINT1 regulatory pathway.

While a number of biomarkers have been identified as having a strong association with preeclampsia (another pregnancy complication associated with placental disease)[19], circulating markers specific for placental insufficiency have been harder to come by, especially during the third trimester. Low circulating PIGF is perhaps the most promising and certainly the most avidly investigated to date[20–22]. Our study suggests circulating SPINT1 around 36 weeks' gestation may have a considerably stronger association with several indicators of placental insufficiency than PIGF.

There will need to be further development of SPINT1 before it can be applied to the clinic, including the generation of a commercial grade ELISA and further validation in other cohorts.

Given circulating SPINT1 may reflect true placental insufficiency, it has the potential to be applied in a variety of other clinical situations to reduce the stillbirth risk, such as monitoring fetuses with co-existing obstetric complications, or those that are post term. Furthermore, combining SPINT1 with other biomarkers (including those that are yet to be reported) or with ultrasound findings at different gestations may yield a highly accurate diagnostic test of fetal growth restriction and placental insufficiency.

A clinical test that helps health care providers better detect fetal growth restriction could reduce the burden of preventable stillbirths, a devastating outcome that prematurely ends around 3 million pregnancies globally each year.

## Methods

**Discovery of new biomarkers of placental insufficiency**. To identify new biomarkers of placental insufficiency we performed the FLAG study, a prospective collection of blood samples from pregnant women at 28 ($27^{+0}$–$29^{+0}$) and 36 ($35^{+0}$–$37^{+0}$ days) weeks gestation from a tertiary referral hospital in Melbourne Australia. This study was approved by the Mercy Health Research Ethics Committee (Ethics Approval Number R14/12) and written informed consent was obtained from all participants.

We divided the cohort approximately in half to discover, and then subsequently validate markers. Samples from the first 1001 consecutively recruited participants constituted cohort 1 and those from the second 1002 consecutively recruited participants constituted cohort 2.

We also recruited a subcohort of 347 nulliparous women in the FLAG study to undergo more intensive studies. They had ultrasound assessments at 36 weeks' gestation to measure blood flow resistance in the uterine, umbilical and the fetal middle cerebral arteries. Where possible, we measured neonatal body composition (lean body mass and fat mass) within 4 days of birth by performing air displacement plethysmography studies using a PEAPOD device. Further methods for the FLAG study are detailed in the Supplementary Information.

We initially screened 22 circulating proteins in a nested case (neonates born small for gestational age, SGA, birthweight <10th centile) control (neonates born >10th centile) set selected from cohort 1. We selected proteins that are highly expressed in placenta relative to other tissues by referencing two bioinformatic databases and choosing proteins that were (1) highly expressed at the mRNA level in placenta relative to all other human tissues (using BioGPS) and (2) its protein product is abundantly expressed on the membrane surface of the placenta (using Protein Atlas).

The proteins screened where circulating concentrations were different among cases of SGA compared to controls were re-assayed in a new batch assay run performed on all samples from cohort 1. Those that remained significantly associated with SGA were then assayed in cohort 2. Further methods on how the proteins were measured are detailed in the Supplementary Information.

**Developing diagnostic tests for placental insufficiency**. Using the data from cohort 2 (the validation run of markers discovered in cohort 1), we examined the

diagnostic performance of potential markers, either alone or in combination, to predict neonates born at birthweight centiles; <20th, <10th, <5th, and <3rd; and birthweight <5th centile but also required nursery admission. We set the specificity at around 90%, which equates to a 10% screen positive rate.

We found SPINT1 performed the best, and no other markers added to its performance. Therefore, we focused on SPINT1. To adjust for the fact that the research ELISA we used exhibited variability in reporting the absolute SPINT1 concentrations between the batch runs for cohorts 1 and 2, we expressed SPINT1 results as multiples of the median (MoMs).

We used cohort 2 (our validation cohort) to develop a 4-tier risk model (that divides the cohort into different levels of SPINT1 MoM concentrations in the circulation at 36 weeks' gestation) that can stratify the entire population into 4 levels of risk of birthing a neonate at a low birthweight.

Further methods describing the statistical analyses are detailed in the Supplementary Information.

**Validation in the MAViS cohort**. To validate the observation that SPINT1 is associated with placental insufficiency we measured SPINT1 in plasma samples obtained from a high-risk cohort in the United Kingdom, the Manchester Antenatal Vascular Service (The MAViS clinic, clinical characteristics shown in Supplementary Table 3 and inclusion/exclusion criteria provided in the Supplementary Methods). Women gave written informed consent to donate samples for future research studies. The study was approved by the NRES Committee North West 11/NW/0426. Women with current hypertension or a hypertensive disorder in a prior pregnancy are referred to the MAViS research clinic from early pregnancy and followed longitudinally with blood samples taken at ~4-week intervals. Such pregnancies are known to have an elevated risk of preeclampsia, SGA or fetal growth restriction. Uterine artery Doppler was performed at each visit and placental size measured at 22–24 weeks gestation by measuring the two widest perpendicular diameters of the placenta. A case-cohort of 291 participants were recruited between October 2011 and December 2016, where a plasma sample was obtained between 24 and 34 weeks and complete outcome data were available were analysed in the current study. The set of 291 participants were selected from a biobank of 518 participants. The clinical characteristics are shown in Supplementary Table 5. The table shows the clinical characteristics of the case-cohort ($n = 291$) selected for this study were no different to the whole consecutive MAViS cohort ($n = 518$). Women attending the MAViS clinic underwent an ultrasound based 'placental screen' at 22–24 weeks gestation as per local guidelines. This consisted of ultrasound assessment of fetal size (Fetal biometry), 2D placental biometry (three dimensions at their widest points), liquor volume (amniotic fluid index and maximum vertical pool depth), uterine artery and umbilical artery Dopplers. Uterine artery was measured by a specialist team of research staff (midwife ultrasound practitioners and Doctors) using standard techniques and an average of triplicate measurements recorded. Placental surface area was calculated by multiplying the two largest measurements from the placenta (width × diameter). Measurements obtained at the same time as the last blood sample are included in the analysis for this study. SPINT1 is expressed a MoMs to correct for gestational changes in SPINT1 in the UK cohort.

**Levels of SPINT1 in preterm growth restriction**. We further investigated SPINT1 by correlating circulating levels with several other clinical indicators of placental insufficiency using data obtained from our subcohort. We also measured circulating SPINT1 concentrations and placental expression in a separate cohort with fetal growth restriction who delivered <34 weeks gestation and in a mouse model of fetal growth restriction[17].

Methods are further described in the Supplementary Information.

**Reporting summary**. Further information on research design is available in the Nature Research Reporting Summary linked to this article.

## Data availability

The source data underlying all figures and supplementary figure are provided as a Source Data file.

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

## Acknowledgements

We thank Sally Beard and Natalie Binder for technical assistance. We thank Emerson Keenan for statistical assistance. We also thank and acknowledge Abigail L. Fowden, Josephine S. Higgins and Owen R. Vaughan for their involvement in design and execution of the mouse studies. We thank Gabrielle Pell, Rachel Murdoch and Genevieve Christophers, Elizabeth Lockie and Emma McLaughlin for their assistance in recruiting and characterizing participants. We acknowledge Alice Robinson for her ultrasound assistance. We also wish to thank the pathology, health information services, and antenatal clinic staff at the Mercy Hospital for Women for their assistance in conducting this research. Funding for this work was provided by the National Health and Medical Research Council (#1065854, #1183854), Foresight Health, The Stillbirth Foundation and the Norman Beischer Medical Research Foundation; Australian Government Research Training Program Scholarship, and RANZCOG Taylor Hammond Scholarship to T.M.M.; National Health and Medical Research Council Fellowships to T.K.L.

(#1159261), S.T. (#1136418), L.H. (#1105603), and N.H. (#1146128). Funding for the mouse work was provided by a PhD studentship, an In Vivo Skills award from the Biotechnology and Biological Sciences Research Council. In addition, the mouse work was supported by a Next Generation fellowship to A.S.P. and PhD studentship from the Centre for Trophoblast Research. The MAViS clinic and research cohort received funding from National Institute Health Research and Tommys Charity and is supported by the Manchester Academic Health Science Centre.

## Author contributions

T.J.K; Performed the bioinformatic studies to select proteins to screen, supervised all laboratory studies, data analysis and wrote the first draft. T.M.M.; Study design, recruitment of patients, ultrasound measures, patient clinical characterisation and data analysis. S.P.W.; Study design, data analysis, overall supervision of the study and manuscript writing. S.T.; Study design, data analysis, overall supervision of the study and wrote the first draft. R.J.H., N.H.; statistical analysis. P.C., T.V.N., R.H., A.H., I.B., N.C., N.J.H., A.S.P.; Performed laboratory studies, including measurement of analytes. K.D., A.M., V.K.; Recruited patients and collected samples. J.M.; Provided samples from the MAVIS cohort from the UK, helped with data analysis and interpretation. L.H.; Helped with study design. N.P.; Helped with clinical characterisation of cohorts. All authors provided input into the final draft of the manuscript.

## Competing interests

T.K.L., T.M.M., S.P.W., and S.T. hold a provisional patent (PCT/AU2019/050516) relating to the use of SPINT1 and syndecan as diagnostic markers in pregnancy. N.H. is the owner of Foresight Health, which has a research and commercialization agreement with The University of Melbourne relating to the development of diagnostic markers of placental insufficiency, and rights to the patent (PCT/AU2019/050516). The remaining authors have no competing interests to declare.
