## [Peer Review File · Nature Communications]

Reviewers' comments:

Reviewer #1 (Remarks to the Author):

SPINT1

This is a study analysing circulating SPINT1 in normal pregnancies and those complicated by growth restriction. SPINT1 is expressed by villous cytotrophoblast and syncytiotrophoblast from early in pregnancy and is responsible for formation of the labyrinth in mice. It is likely to have role in human placental development

The use of the term 'placental insufficiency' is outdated as used in this report. Severe FGR arises because of failure of normal development of the placenta in the first trimester. The normal villous tree fails to develop properly meaning the surface area for gas exchange and nutrients is less.

Functional studies have been performed with primary term trophoblast cells and HTR8 cell line. Neither of these are optimal. Given that the origin of the FGR in early in development, and that relative hypoxia is secondary to the primary defect, the use of term trophoblast cells is problematic. Now available are 2D and 3D trophoblast stem cell lines derived from blastocysts and first trimester placentae. These should be used and perhaps a more relevant end-point such a proliferation.

This has been tested with the HTR8 cell line. It is uncertain if this line is trophoblast. If it is it is generally considered as extravillous trophoblast. The expression of HLA-G on the line in use should be checked as it is also notoriously variable. SPINT1 is predominantly expressed by the proliferative villous cells in first trimester; it is downregulated in EVT.

Given these problems and the availability of bone fide first trimester trophoblast cells (Okoe 2018, Haider 2018, Turco 2018), it seems better to drop this part of the manuscript and analyse function of SPINT1 in depth with these systems.

Reviewer #2 (Remarks to the Author):

This paper evaluated potential biomarkers for placental insufficiency based on prospective collected plasma samples at 28 and 36 weeks of pregnancy. Circulating SPINT1 was discovered in a cohort of 997 individuals from the FLAG study and then validated in a cohort of 999 from the same study as well as

another independent cohort from the United Kingdom based on associations with several clinical indicators of placental insufficiency. The research addressed in this paper is important and the paper is in general well-written. I have a few major comments.

In the introduction the current clinical approach of using ultrasound to detect SGA was criticized for its low sensitivity (20-30%) and the concern that SGA itself may not be a good indicator of placental insufficiency. Yet the primary analysis for discovering and validating biomarkers is based on their associations with SGA. In Table 1, for birthweight < 10th percentile, SPINT1 MoM<0.63 has a sensitivity around 19.0%. How do we know that the new test is better than clinical approach or offer additional value when combined with clinical approach?

From supplementary table S1/S2, there are other participant characteristics related to SGA, e.g. smoking status. Have the authors evaluated predictive model by including those participant characteristics and see whether the biomarker offers additional value?

One objective of cohort 2 was to try to find a combination of markers. It was briefly mentioned in page 11 that using logistic regression approaches the authors were unable to identify a multi-marker predictive test that added to the performance of SPINT1 alone without presenting additional details. It would be useful to provide some results showing how the conclusion was reached.

Minor Comments:

The supplementary material needs better arrangement (e.g. Supplementary S2)

Figure caption (Figure 1): AUC should be spelled as "Area under the ROC curve". Also it would be useful to provide confidence interval for AUC in figures

Table 1 captions mentions cohort 2 has 997 participants (instead of 999)

Reviewer #3 (Remarks to the Author):

The manuscript reports the results of a prospective study in which maternal plasma was sampled at 28 and 36 weeks of gestation to identify biomarkers of placental insufficiency related to small for gestational age neonates diagnosed at delivery. The authors divided the cohort into discovery and validation sets.

The investigators screened for 22 proteins in cohort 1, and identified a subset subsequently tested in the validation cohort. It was shown that SPINT1 at 28 weeks of gestation is decreased in women who will deliver an SGA neonate, and that this marker measured at 36 weeks of gestation is associated with several clinical indicators of placental insufficiency.

The authors performed in-vitro functional studies that support their claim. Could the authors consider the following:

1) It is not clear whether the transformation of the SPINT1 and other markers (e.g. PIGF) into MoM values included adjustment for relevant maternal characteristics. This would be important since factors such as smoking, parity, maternal weight (some of which are different between cases and controls, or have a trend in that direction) are also known to affect the levels of biomarkers, and, hence, are typically included in MoM calculations in other studies (see for instance [https://www.ajog.org/article/S0002-9378\(15\)02345-5/fulltext](https://www.ajog.org/article/S0002-9378(15)02345-5/fulltext)).

2) Relative risk estimates for a low SPINT1 at 36 weeks gestation are rather similar to those of PIGF (Table 1 vs Table S3); yet, it is not clear whether the two markers identify the same women as being at risk. A figure showing the correlation between SPINT1 and PLGF would be informative.

3) The PLGF/sFLT1 ratio (also called Angiogenic Index 1) is a well now marker for placental dysfunction (PMID: 26688491). The authors could report the prediction performance for the PLGF/sFLT1 ratio, as was done for PIGF in Table S3.

4) The MoM data reported in the figures could perhaps be represented using a logarithmic scale to enhance visualization.

5) The fundamental question is whether this new marker can outperform PIGF, but this is an important contribution, and the study has been well-conducted and addresses an important healthcare issue. Some of these matters are addressed on page 14, when the authors compare the performance of SPINT1 vs PIGF on birthweight placental, lean mass, and uterine artery pulsatility.

Reviewer #1 (Remarks to the Author):

1) This is a study analysing circulating SPINT1 in normal pregnancies and those complicated by growth restriction. SPINT1 is expressed by villous cytotrophoblast and syncytiotrophoblast from early in pregnancy and is responsible for formation of the labyrinth in mice. It is likely to have role in human placental development

The use of the term 'placental insufficiency' is outdated as used in this report. Severe FGR arises because of failure of normal development of the placenta in the first trimester. The normal villous tree fails to develop properly meaning the surface area for gas exchange and nutrients is less.

Response: We would be keen to continue to use the term placental insufficiency given that it is widely used in the literature and in contemporary clinical guidelines. In our opinion, irrespective of the gestation, the term 'placental insufficiency' remains valid. It is an apt description that describes the pathophysiology, ie. where the placenta is unable to meet the oxygen and nutrient requirements of the fetus to maintain normal growth and development.

The term 'Placental insufficiency' is widely used in the contemporary literature, evidenced by:

- A pubmed search of "placental" AND "insufficiency" over the last 2 years alone yields 276 articles (24/10/2017 until 24/10/2019).
- It is frequently used throughout influential clinical guidelines published by the Royal College of Obstetrics and Gynaecology (Greentop Guidelines, 2014 freely downloadable) and American College of Obstetricians and Gynaecologists (ACOG guidelines, Obstet Gynecol Feb 2019), and other international guidelines.
- Reviewers 2 and 3 have used this term in their report implying they consider its use acceptable.

Furthermore, whilst the origin of preterm FGR may be in the first trimester, this is not the case for all cases of FGR, especially for late preterm and term disease (the focus of our work) where it is plausible that biological events occurring well beyond the first trimester may be responsible for a decreased growth trajectory across the last trimester.

We refer the reviewer to the ACOG guidelines (Obstet Gynecol Feb 2019) that summarise the multiple aetiologies that "lead to the same final common pathway of decreased fetal nutrition".

2) Functional studies have been performed with primary term trophoblast cells and HTR8 cell line. Neither of these are optimal. Given that the origin of the FGR in early in development, and that relative hypoxia is secondary to the primary defect, the use of term trophoblast cells is problematic. Now available are 2D and 3D trophoblast stem cell lines derived from blastocysts and first trimester placentae. These should be used and perhaps a more relevant end-point such a proliferation.

This has been tested with the HTR8 cell line. It is uncertain if this line is trophoblast. If it is it is generally considered as extravillous trophoblast. The expression of HLA-G on the line in use should be checked as it is also notoriously variable. SPINT1 is predominantly expressed by the proliferative villous cells in first trimester; it is downregulated in EVT.

Given these problems and the availability of bone fide first trimester trophoblast cells (Okae 2018, Haider 2018, Turco 2018), it seems better to drop this part of the manuscript and analyse function of SPINT1 in depth with these systems.

Response: We wish to firstly clarify that the discovery and validation of low circulating SPINT1 as a potential clinical biomarker of placental insufficiency is the main finding of this

report. And our most novel finding. We do not propose that SPINT1 has a role during the first trimester causing the defects that lead to FGR.

1) Established FGR is associated with chronic placental hypoxia. Our primary placental cell experiments confirm that hypoxic exposure decreases intracellular and secreted SPINT1 concentrations. We suggest that this provides valuable functional information for the reader, and thus should be retained in the manuscript. These experiments suggest that placental hypoxia may be responsible for driving the decrease in circulating SPINT1 seen in FGR (even if the placental hypoxia arises secondary to a primary defect).

2) As requested, we have removed the data showing experiments performed in cell lines (Fig 4J and K of the original submission) in light of the reviewer's concerns that the placental cell origin they represent is unclear.

We do not believe removing this data diminishes the impact of our primary finding – ie the diagnostic potential of SPINT1 as a circulating biomarker. We still present a strong body of evidence linking low SPINT1 with placental insufficiency: human correlative studies with multiple clinical parameters associated with placental insufficiency (in both the Australian and UK cohorts), in vitro human placental studies (primary cells), and in vivo mouse studies (see Figs 3e,f,g, and Figs 4a-i)

3) The 2D and 3D assays suggested would indeed provide new insights into placental development. However, these assays are beyond the scope of this report as our primary focus was not to investigate the role for SPINT1 in first trimester placental development.

Reviewer #2 (Remarks to the Author):

This paper evaluated potential biomarkers for placental insufficiency based on prospective collected plasma samples at 28 and 36 weeks of pregnancy. Circulating SPINT1 was discovered in a cohort of 997 individuals from the FLAG study and then validated in a cohort of 999 from the same study as well as another independent cohort from the United Kingdom based on associations with several clinical indicators of placental insufficiency. The research addressed in this paper is important and the paper is in general well-written.

Response: We thank the reviewer for these comments.

I have a few major comments.

1) In the introduction the current clinical approach of using ultrasound to detect SGA was criticized for its low sensitivity (20-30%) and the concern that SGA itself may not be a good indicator of placental insufficiency. Yet the primary analysis for discovering and validating biomarkers is based on their associates with SGA. In Table 1, for birthweight < 10th percentile, SPINT1 MoM<0.63 has a sensitivity around 19.0%. How do we know that the new test is better than clinical approach or offer additional value when combined with clinical approach?

Response: It is not possible to be certain from our current study whether the measurement of circulating SPINT1 would improve upon current clinical methods for the detection of SGA. This could only be addressed with a prospective study. However, we would like to offer some observations:

- The current clinical approach (measurement of the abdomen with a tape measure followed by selective ultrasound) could not provide accuracy comparable to a 4 tier risk stratification model such as we have proposed in table 2. The reason is that there is obvious way that information obtained from the tape measure can stratify risk in this way.

- It is known that amongst small for gestational age fetuses (<10th birthweight centile) there will be babies that are constitutionally small and healthy, and others affected by placental dysfunction/insufficiency who have a higher risk of stillbirth. A test that selectively identifies those affected by true placental insufficiency rather than simply identifying smallness would be the most desirable test to discover.

Thus, the exciting possibility stemming from our work is that we have presented data suggesting low SPINT1 may be specifically associated with true placental insufficiency. Our findings suggest that low circulating SPINT1 could identify fetuses affected by placental insufficiency among all those born <10th centile birthweight. If confirmed in prospective studies this would transform obstetric care.

We also wish to note that we have replaced table 1 of the current submission with the table that follows.

SPINT1 diagnostic performance:

	Birthweight < 3rd centile	Birthweight < 5th centile	Birthweight < 10th centile	Birthweight < 20th centile	Birthweight < 5rd centile and nursery admission
Positive Predictive Value	10.2% (5.2 – 17.5)	16.1% (9.8 – 24.2)	22.6% (15.3 – 31.3)	37.9% (29.3 – 47.1)	4.8% (1.6 – 10.8)
Negative Predictive Value	98.3% (97.2 – 99.1)	96.5% (95.1 – 97.6)	90.9% (88.9 – 92.8)	78.4% (75.5– 81.1)	99.7% (99.0 – 99.9)
Risk Ratio (95%CI)	6.04 (2.85 – 12.82)	4.59 (2.66 – 7.93)	2.50 (1.68 – 3.71)	1.75 (1.35 – 2.27)	14.2 (3.44 – 58.47)
Sensitivity	42.3% (23.4 – 63.1)	36.7% (23.4 – 51.7)	24.5% (16.7 – 33.8)	19.9% (15.0 – 25.6)	62.5% (24.5 – 91.5)
Specificity	90.0% (88.0 – 91.8)	90.1% (88.0 – 91.9)	90.0% (87.9 – 91.9)	89.9% (87.5 – 91.9)	89.9% (87.9 – 91.7)

The minor difference is that table 1 of original submission used a SPINT1 MoM<0.63 cut off so that it was consistent with the first tier of risk shown in table 2 (ie the 4 tier risk stratification model that we have proposed). However, examining SPINT1 on its own as a rule in/rule out test (without tiers), has slightly improved performance. *This means the sensitivity for <10th birthweight centile is now 24.5% instead of 19%.*

2) From supplementary table S1/S2, there are other participant characteristics related to SGA, e.g. smoking status. Have the authors evaluated predictive model by including those participant characteristics and see whether the biomarker offers additional value?

Response: 1) We have now undertaken a new analysis examining the predictive ability of circulating SPINT1 at 36 weeks' gestation to detect fetuses <10th, 5th and 3rd centiles, adjusting for age, smoking, parity and BMI.

Adding these characteristics did not improve (nor hinder) the performance of SPINT1, as shown below.

Model	ROC area (95%CI)	Sensitivity at 90% specificity	LR +	LR -
<10th centile:				
spint1 alone	0.66 (0.61 – 0.72)	28.7	3.04	0.79
spint1 adjusted for clinical factors	0.66 (0.61 – 0.72)	27.6	2.97	0.80

<5th centile:				
spint1 alone	0.71 (0.62 – 0.79)	27.6	3.38	0.79
spint1 adjusted for clinical factors	0.70 (0.62 – 0.79)	27.7	2.38	0.81
<3th centile:				
spint1 alone	0.75 (0.64 – 0.85)	35.4	4.60	0.70
spint1 adjusted for clinical factors	0.76 (0.67 – 0.85)	43.7	5.24	0.61

Clinical factors adjusted for were age, smoking, parity and Body mass Index. Analysis was performed using logistic regression analysis. ROC – Receiver Operated Curve. CI – Confidence Interval. LR – Likelihood ratio. Analysis was done on cohort 2. Total numbers where all clinical variables were available for this analysis were as follows: n=959 for <10th centile analysis; n=963 for <5th and <3rd centile analyses. N numbers vary to those presented in table 1 (n=998) because of missing maternal characteristics for some patients.

This further supports the contention that SPINT1 is the final downstream signal of placental insufficiency, irrespective of the contributory risk factors or comorbidities.

We have added this analysis in the paper. We have referred to this in the results section of the manuscript and added as supplementary figure table S5.

2) We also performed further analyses where we combined multiple markers alone (SPINT/PIGF/GDF15/Syndecan, table 1 below), OR combined these multiple biomarkers PLUS adjusted the analysis for age, smoking, parity and BMI (table 2 below). We used two statistical approaches: logistic and LASSO penalized regression.

Adjusting for these clinical parameters did not impact on the performance of the multi-marker tests to predict pregnancies with a birthweight <10th centile (compare table 1 [multiple marker modelling] vs table 2 [multiple marker modelling adjusting for clinical characteristics]). None of these models outperformed SPINT1 alone.

Table 1: Modelling by combining multiple biomarkers (without correcting for maternal factors)

Model	ROC area (95%CI)	Sensitivity at 90% specificity	LR +	LR -
Logistic regression				
Spint1 + GDF15 + PIGF + Syndecan	0.67 (0.62 – 0.73)	27.8	3.00	0.80
Spint1 + GDF15	0.66 (0.61 – 0.72)	28.6	3.09	0.79
Spint1	0.66 (0.61 – 0.72)	28.7	3.04	0.79
GDF15	0.60 (0.54 – 0.65)	19.2	2.07	0.89
PLGF	0.60 (0.54 – 0.66)	13.6	1.46	0.96
Syndecan	0.61 (0.55 – 0.67)	21.0	2.27	0.87
Lasso¹				
SPINT1 + GDF15 + PLGF + Syndecan	0.68 (0.63 – 0.73)	28.8	3.06	0.79

¹ lambda = 0.0018269, no variable dropped. Total numbers where all clinical variables and biomarker values were available for this analysis was 959 out of the set of 999 in cohort 2.

Table 2: Modelling by combining multiple biomarkers AND adjusting for maternal Age, smoking, Parity and body mass index.

Model	ROC area (95%CI)	Sensitivity at 90% specificity	LR +	LR -
Logistic regression				
Spint1 + GDF15 + PIGF + Syndecan	0.67 (0.62 – 0.73)	27.8	3.00	0.80
Spint1 + GDF15	0.66 (0.61 – 0.72)	28.6	3.09	0.79
Spint1	0.66 (0.61 – 0.72)	27.6	2.97	0.80
GDF15	0.60 (0.54 – 0.65)	19.2	2.07	0.89
PLGF	0.60 (0.54 – 0.66)	13.6	1.46	0.96
Syndecan	0.61 (0.55 – 0.67)	21.0	2.27	0.87
Lasso ¹				
SPINT1 + GDF15 = PLGF + Syndecan	0.68 (0.63 – 0.73)	28.8	3.06	0.79

¹ lambda = 0.0041783, final model dropped BMI, parity and smoking. Total numbers where all clinical variables and biomarker values were available for this analysis was 959 out of the set of 999 in cohort 2.

There were too few cases to perform similar modelling to predict birthweights <5th and <3rd centiles (ie we were underpowered to perform multi-marker modelling and also adjust for 4 maternal characteristics).

We have already added the data to confirm that inclusion of clinical characteristics with SPINT1 did not affect the diagnostic performance (Supplementary table S5 which is the table presented in response 1 to this reviewer). We feel that these new analyses relating to *multi*-markers AND adjusting for maternal characteristics should not be included in the manuscript given that 1) we could only perform this analysis for <10th centile (which was not significantly better than SPINT1, as seen when comparing the two tables above) and 2) we were unable to do similar modelling for <5th and <3rd centiles as there were too few cases.

3) One objective of cohort 2 was to try to find a combination of markers. It was briefly mentioned in page 11 that using logistic regression approaches the authors were unable to identify a multi-marker predictive test that added to the performance of SPINT1 alone without presenting additional details. It would be useful to provide some results showing how the conclusion was reached.

Response: As requested we have now included three new tables (supplementary tables S6, S7 and S8) that present the diagnostic performance of circulating SPINT1 alone, vs combinations of other markers with SPINT1 (PIGF/GDF15/syndecan).

The tables show the diagnostic performance of SPINT1 alone is not improved when combined with other markers to predict a birthweight <10th, 5th or 3rd centile.

Minor Comments:

The supplementary material needs better arrangement (e.g. Supplementary S2)

Response: We have significantly reworked the supplementary materials sections and hope it is more easily followed.

Figure caption (Figure 1): AUC should be spelled as "Area under the ROC curve". Also it would be useful to provide confidence interval for AUC in figures

Response: "Area under the ROC curve" corrected.

95% confidence intervals for the ROCs have now been incorporated into all figures.

Table 1 captions mentions cohort 2 has 997 participants (instead of 999)

Response: Corrected, with thanks to 998.

Reviewer #3 (Remarks to the Author):

The manuscript reports the results of a prospective study in which maternal plasma was sampled at 28 and 36 weeks of gestation to identify biomarkers of placental insufficiency related to small for gestational age neonates diagnosed at delivery. The authors divided the cohort into discovery and validation sets.

The investigators screened for 22 proteins in cohort 1, and identified a subset subsequently tested in the validation cohort. It was shown that SPINT1 at 28 weeks of gestation is decreased in women who will deliver an SGA neonate, and that this marker measured at 36 weeks of gestation is associated with several clinical indicators of placental insufficiency. The authors performed in-vitro functional studies that support their claim. Could the authors consider the following:

1) It is not clear whether the transformation of the SPINT1 and other markers (e.g. PIGF) into MoM values included adjustment for relevant maternal characteristics. This would be important since factors such as smoking, parity, maternal weight (some of which are different between cases and controls, or have a trend in that direction) are also known to affect the levels of biomarkers, and, hence, are typically included in MoM calculations in other studies (see for instance [https://www.ajog.org/article/S0002-9378\(15\)02345-5/fulltext](https://www.ajog.org/article/S0002-9378(15)02345-5/fulltext)).

Response: In our study, the purpose of expressing SPINT1 as MoMs was to adjust for gestational changes in SPINT1 in the UK cohort. In the cohorts investigated there was not a significant association between weight, smoking status or parity (see Supplementary Table S10) and SPINT1 and therefore we did not have to adjust for these characteristics. We have clarified the reason for expressing SPINT1 as MoMs in the methods.

We have now performed new analyses' where we have adjusted for maternal characteristics. Please see our response to point 2, reviewer 2.

2) Relative risk estimates for a low SPINT1 at 36 weeks gestation are rather similar to those of PIGF (Table 1 vs Table S3); yet, it is not clear whether the two markers identify the same women as being at risk. A figure showing the correlation between SPINT1 and PLGF would be informative.

AND

5) The fundamental question is whether this new marker can outperform PIGF, but this is an important contribution, and the study has been well-conducted and addresses an important healthcare issue. Some of these matters are addressed on page 14, when the authors compare the performance of SPINT1 vs PIGF on birthweight placental, lean mass, and uterine artery pulsatility.

Response: PIGF alone or in combination with sFlt1 as a ratio is currently the best reported circulating marker for placental insufficiency and FGR. The question raised by the reviewer

(Q 2 & 5) asks whether there is evidence that SPINT1 outperforms PIGF as a marker of placental insufficiency.

Point 2 suggests a correlative analysis between SPINT1 and PIGF. Unsurprisingly, there is indeed a strong correlation between these analytes ($p < 0.0001$ and $r = 0.14$ (95% CI 0.08-0.20)). However, any correlative analysis is likely to be significant for any analyte we find to be differentially expressed with low birthweight. Such a simple correlation has perhaps limited value in compare the relative diagnostic performance of these markers.

What may be a more informative comparison is to statistically compare the area under the ROC curves. The following analysis statistically compares the area under the ROC for SPINT1 vs PIGF to detect fetuses with a birthweight $< 10^{\text{th}}$ centile in cohort 2.

A comparison of the ROC curves generated from SPINT1 and PIGF (DeLong test statistic) reveals a statistically significant difference ($p = 0.03$):

We have added this new analysis, referring to it in the results section of the manuscript and we have added this to figure 1 of the main manuscript (Fig 1g). We have added this to the main manuscript given the understandable interest in comparing our marker to PIGF.

Our manuscript now presents a strong case that circulating SPINT1 has a stronger association with placental insufficiency than PIGF, evidenced by:

- 1) Statistical difference when comparing area under the ROC (as above)
- 2) SPINT1 has a stronger level of statistical significance in cohort 2 compared with PIGF (Compare Figs 1c,d vs 1e,f).
- 3) SPINT1 had stronger associations with multiple clinical parameters of placental insufficiency compared with PIGF (see supplementary Figure S5).

3) The PLGF/sFLT1 ratio (also called Angiogenic Index 1) is a well now marker for placental dysfunction (PMID: 26688491). The authors could report the prediction performance for the PLGF/sFLT1 ratio, as was done for PIGF in Table S3.

Response: As requested we have generated the following table that shows the performance of the sFlt/PIGF ratio:

Diagnostic performance of the sFlt/PIGF ratio in cohort 2.

	Birthweight < 3rd centile	Birthweight < 5th centile	Birthweight < 10th centile	Birthweight < 20th centile	Birthweight < 5rd centile and nursery admission
Positive Predictive Value	4.0%	8.7%	14.4%	24.3	1.0%
Negative Predictive Value	97.5%	95.5%	89.8%	76.4	99.2%
Risk Ratio (95%CI)	1.61 (0.57 – 4.59)	1.96 (0.98 – 3.91)	1.56 (0.94 – 2.59)	1.02 (0.72 – 1.48)	1.27 (0.16 – 10.21)
Sensitivity	15.4%	18.4%	14.2%	10.6%	12.5%
Specificity	90.0%	90.1%	90.0%	89.8%	89.9%

Expressing the data as a PIGF/sFlt ratio provides identical performance characteristics.

In our cohorts, the predictive performance when sFlt was added to PIGF as a ratio was inferior to PIGF alone (supplementary table S4) and significantly inferior to the performance of SPINT1. This is not surprising given sFlt1 alone did not have a significant association with birthweight <10th centile in cohort 2.

We have added this table to the manuscript, supplementary table S9. It is referred to in the results section.

We are also aware that while many investigators have expressed the ratio as an sFlt/PIGF ratio, there are some published reports where the ratio is presented inversely (ie PIGF/sFlt, see PMID: 26688491, cited by the reviewer). We therefore also performed the analysis as a PIGF/sFlt ratio and the results were identical. We have included this point at the bottom of the table we have now included in the modified manuscript.

4) The MoM data reported in the figures could perhaps be represented using a logarithmic scale to enhance visualization.

Response: We agree that utilising a logarithmic scale can sometimes add clarity to figures, but mainly in situations where the data is skewed. Our data is not particularly skewed and the datapoints seem quite easily visualised (see Figures 2 and 3 where MoM data is presented).

Below are representative data from figure 3 with the logarithmic scale, or not (original figure). We do not think the logarithmic scale adds clarity so we have left it unaltered; we would be happy to change this at the Editor's request.

The figure on the left is Fig 3a of the manuscript; figure the right is the same data with a log scale

Figure on the left is Fig 3c of the manuscript; the figure on the right is the same data with a log scale.

Reviewers' comments:

Reviewer #1 (Remarks to the Author):

The authors have responded to the points raised and this is an important contribution to the clinical assessment of fetal growth at 36 weeks gestation.

I remain of the view that 'placental insufficiency' is an unhelpful term but accept it is widely used by clinicians when the underlying biological mechanisms are unexplained.

To quote the definitive book on the placenta, Bernischke, p622:

" the term placental insufficiency subtly implies that there is an intrinsic defect in placental function or its development. We believe that the concept of 'placental insufficiency' to be misleading and one that diverts attention from the search of pathogenetic mechanisms".

The term 'hypoxia' is also problematic and needs clarifying. The mother is not hypoxic. There may be uteroplacental hypoxia because the arterial supply is inadequate later in gestation. Or there may be impaired oxygen exchange and the mother is normoxic but the fetus is hypoxic. Hypoxia is also a state depending on the tissue's metabolic requirements. The mechanisms underlying low SPINT1 levels in reduced fetal growth are not clear.

The paper stands on its own without introducing one small experiment using human term cytotrophoblast. To work out where and when in gestation the production of SPINT1 is reduced will require detailed functional investigations that are not the focus of this report. It is assumed that the intervillous space is hypoxic but, see above, there could be other mechanisms. Is there reduced protein synthesis in trophoblast because of reperfusion injury? Is there failure to grow the normal villous tree? Is there a failure of syncytialization? Are there non-trophoblast sources of SPINT1?

The murine model does back their findings up but the human experiment seems inconclusive and I still believe does not add to their important findings.

Reviewer #2 (Remarks to the Author):

The authors have adequately addressed most of my previous comments. I only have one minor comment below:

Previously I asked about whether the biomarker adds additional values to other participant characteristics related to SGA (e.g. age, smoking, etc). The authors have responded by comparing the model with both biomarker and clinical factors and the model with biomarker alone, and shown that clinical factors did not add much after biomarker is included. I am actually more interested in seeing performance of the model with clinical factors alone and seeing whether adding SPRINT1 further improve the model given that clinical factors are easier to obtain than biomarker.

Reviewer #3 (Remarks to the Author):

The authors have amended the manuscript appropriately and I believe it is ready for publication.

January 16th, 2020

We thank the reviewers for their comments and provide a point-by-point response to each of their comments.

Kind regards,

Associate Professor Tu'uhevaha Kaitu'u-Lino, corresponding author

Reviewer #1 (Remarks to the Author):

The authors have responded to the points raised and this is an important contribution to the clinical assessment of fetal growth at 36 weeks gestation.

I remain of the view that 'placental insufficiency' is an unhelpful term but accept it is widely used by clinicians when the underlying biological mechanisms are unexplained.

To quote the definitive book on the placenta, Bernischke, p622: " the term placental insufficiency subtly implies that there is an intrinsic defect in placental function or its development. We believe that the concept of 'placental insufficiency' to be misleading and one that diverts attention from the search of pathogenetic mechanisms".

Response: We thank the reviewer for this thoughtful comment and take on board the view that the term 'placental insufficiency' is not universally accepted. However, we still prefer to use the term 'placental insufficiency' given it is so widely used. We addressed this point in more detail in our prior rebuttal.

The term 'hypoxia' is also problematic and needs clarifying. The mother is not hypoxic. There may be uteroplacental hypoxia because the arterial supply is inadequate later in gestation. Or there may be impaired oxygen exchange and the mother is normoxic but the fetus is hypoxic. Hypoxia is also a state depending on the tissue's metabolic requirements. The mechanisms underlying low SPINT1 levels in reduced fetal growth are not clear.

Response:

- 1) We have removed the experiment where we exposed cells to hypoxia as an in vitro model of placental insufficiency (in light of reviewer 1's comment following this one). This now removes most of the discourse related to the term 'hypoxia'.**
- 2) The murine model is now the only experiment where hypoxia was used (chronic maternal hypoxia to induce a fetal growth restriction model). We have added the following text in the discussion to acknowledge the reviewer's concern about this model.**

"To induce fetal growth restriction in our murine model we induced maternal hypoxia. Our model is limited because when there is fetal growth restriction in humans the mother is not hypoxic, meaning our in vivo model does not entirely mimic the pathophysiology of this disease. Nevertheless, this model is still likely to induce fetal hypoxia (which does occur with fetal growth restriction) and is therefore still relevant to human disease."

The paper stands on its own without introducing one small experiment using human term cytotrophoblast. To work out where and when in gestation the production of SPINT1 is reduced will require detailed functional investigations that are not the focus of this report. It is assumed that the intervillous space is hypoxic but, see above, there could be other mechanisms. Is there reduced protein synthesis in trophoblast because of reperfusion injury? Is there failure to grow the normal villous tree? Is there a failure of syncytialization? Are there non-trophoblast sources of SPINT1?

The murine model does back their findings up but the human experiment seems inconclusive and I still believe does not add to their important findings.

Response: As requested we have moved the experiments using term cytotrophoblast ie Figures 4e-g in the prior draft. We have reworked the manuscript accordingly.

Reviewer #2 (Remarks to the Author):

The authors have adequately addressed most of my previous comments. I only have one minor comment below:

Previously I asked about whether the biomarker adds additional values to other participant characteristics related to SGA (e.g. age, smoking, etc). The authors have responded by comparing the model with both biomarker and clinical factors and the model with biomarker alone and shown that clinical factors did not add much after biomarker is included. I am actually more interested in seeing performance of the model with clinical factors alone and seeing whether adding SPINT1 further improve the model given that clinical factors are easier to obtain than biomarker.

Response: As requested we have performed new analyses to examine the performance of clinical factors alone in predicting low birthweight <10th, 5th and 3rd centiles.

These analyses confirm circulating SPINT1 alone at 36 weeks' gestation outperforms combining clinical factors to predict birthweight <10th, 5th and <3rd (see the additions to Supplementary Table 5 highlighted in yellow, and compare their performance to SPINT1 ALONE).

We have added this data to supplementary table S5.

Model (n = 967)	ROC area (95%CI)	Sensitivity at 90% specificity	LR +	LR -
<10th centile birthweight:				
Spint1 alone	0.66 (0.61 – 0.72)	24.5	2.46	0.84
Spint1 adjusted for clinical factors	0.66 (0.61 – 0.72)	22.7	2.26	0.86
Clinical factors only	0.57 (0.51 – 0.63)	14.4	1.44	0.95
<5th centile birthweight:				
Spint1 alone	0.71 (0.62 – 0.79)	36.7	3.71	0.70
Spint1 adjusted for clinical factors	0.70 (0.62 – 0.79)	37.2	3.74	0.70

Clinical factors only	0.64 (0.55 – 0.72)	23.3	2.33	0.85
<3th centile birthweight:				
Spint1 alone	0.75 (0.64 – 0.85)	42.3	4.23	0.64
Spint1 adjusted for clinical factors	0.76 (0.67 – 0.85)	41.7	4.18	0.65
Clinical factors only	0.69 (0.59 – 0.78)	25.0	2.51	0.83

Supplementary Table 5: Diagnostic performance of circulating SPINT1 at 36 weeks' gestation to predict <10th, 5th and <3rd centile birthweights with adjustments for maternal clinical factors. Clinical factors adjusted for were age, smoking, parity and Body Mass Index. Analysis was performed using logistic regression analysis. Analysis was done on cohort 2. Total numbers where all clinical variables were available for analyses were n = 967. Numbers vary to those presented in table 1 (n=998) because of missing maternal characteristics for some patients. ROC – Receiver Operated Curve. CI – Confidence Interval. LR – Likelihood ratio.

We have added the following text to the main manuscript at the end of the subsection of results with the subheading 'Developing validating diagnostic tests to predict low birthweight'

SPINT1 also performed better as a diagnostic test compared to combining clinical characteristics that are associated with an increased risk of birthing a low birthweight neonate (Supplementary Table S5).

Reviewer #3 (Remarks to the Author):

The authors have amended the manuscript appropriately and I believe it is ready for publication.

REVIEWERS' COMMENTS:

Reviewer #2 (Remarks to the Author):

The authors have adequately addressed my questions.

March 11th 2020

Dear Editors of Nature Communications,

We are pleased to provide a point-by-point response to each of the Reviewer comments.

Regards,

Tu'uhevaha Kaitu'u-Lino, corresponding author

Reviewer comments:

Reviewer #2 (Remarks to the Author):

The authors have adequately addressed my questions.

We thank reviewer 2 for this response.